# Poverty, quality of life and psychological wellbeing in adults with congenital heart disease in Chile

**Rodrigo López Barreda**[1]☉*, **Alonso Guerrero**[2]☉, **Juan Cristóbal de la Cuadra**[2]☉, **Manuela Scotoni**[3]☉, **Wilbaldo Salas**[1]‡, **Fernando Baraona**[4,5]‡, **Francisca Arancibia**[5,6]‡, **Polentzi Uriarte**[5]‡

**1** Anesthesiology Department, School of Medicine, Pontificia Universidad Católica de Chile, Santiago, Chile, **2** School of Medicine, Pontificia Universidad Católica de Chile, Santiago, Chile, **3** Anesthesiology Department, Kantonsspital Winterthur, Winterthur, Switzerland, **4** Cardiology Department, School of Medicine, Pontificia Universidad Católica de Chile, Santiago, Chile, **5** Cardiology Department, Instituto Nacional del Tórax, Santiago, Chile, **6** Pediatrics Department, School of Medicine, Pontificia Universidad Católica de Chile, Santiago, Chile

☉ These authors contributed equally to this work.
‡ WS, FB, FA and PU also contributed equally to this work.
* ralopez@uc.cl

**Data Availability Statement:** Data are available upon request for researchers who meet the criteria for access to confidential data. As the population is very particular (adult patients suffering from

## Abstract

The objective of this study was to assess the quality of life and psychological wellbeing of adults with congenital heart disease (CHD) in Chile, and to identify other associated factors. The study enrolled 68 patients aged between 18 and 72 (median 29), 35 being females. They completed a questionnaire, which included a quality of life assessment tool (the Medical Outcome Study 36-Item Short Form Health Survey), a number of psychological scales (the General Health Questionnaire, the Basic Psychological Needs Scales and the Beck Hopelessness Scale), a socioeconomic survey, and some clinical data. CHD patients reported worse scores in those scales assessing physical dimensions of quality of life (physical function (70.5), physical role functioning (64), vitality (65.3)), and general quality of life (58.6), than in emotional or social dimensions. Female gender was associated with lower scores in physical function (59.12 versus 82.66; p<0.01) and physical role functioning (53.68 versus 75; p<0.05); poverty was associated with worse results in physical function (61.92 versus 82.96; p<0.01), role physical (53.21 versus 79.63; p<0.01), vitality (60.89 versus 71.67; p<0.05), social role functioning (70.19 versus 82.87; p<0.05) and bodily pain (65.77 versus 81.2; p<0.05). Furthermore, we found that psychological scales had an association with quality of life, but clinical variables did not show significant correlations to any dimension. Poverty has an impact on the quality of life of CHD patients. This population only has a decrease in the quality of life physical dimensions, suggesting that quality of life depends on many different factors.

congenital heart diseases and being followed in a particular hospital), even if personal information was deleted, the participants would be easily identifiable; moreover, the scales used in this study collect information that has been considered highly sensitive. Therefore, there are ethical reasons that impede the publication of our data set. The IRB that assessed this study and imposed this restriction is the Comité Ético Científico de Ciencias de la Salud. Data access requests may be made to the IRB at its email: eticadeinvestigacion@uc.cl or its phone: (+56) 223548173 / (+56) 22354-2397

**Funding:** The author received no specific funding for this work.

**Competing interests:** The authors have declared that no competing interests exist.

## Introduction

Congenital Heart Diseases (CHD) include a wide range of cardiac disorders that are present at birth, ranging from simple conditions to severe structural abnormalities. It is estimated that approximately 1% of children are born with CHD and a number of them die during the first year of life [1]. However, mortality has sharply decreased in the last decades thanks to improvements in technology and medical care. Nowadays, more than 85% of these patients reach adulthood [2]. Healthcare for these patients has become a challenge, not only because of medical issues, but also because patients have quality of life (QoL) and psychological functioning needs that should be met [3]. For a better understanding of the impact of disorder and therapy on patients' lives, surveying QoL and psychological functioning is essential [4].

For this reason, the QoL of CHD patients has been extensively surveyed. Remarkably, in this regard, the literature is mostly focused on clinical features and shows ambiguous results. While some studies in children and adolescents suffering from CHD showed a decrease in QoL [5, 6], others found these trends only in specific life domains [5, 7], and other reports claimed that patients with CHD enjoy a better QoL than the general population [8–10]. It is not surprising that studies aiming at identifying those patients at risk for low QoL provided contradictory results. Older age has been identified as a risk factor [11–13], but Silva found the opposite [9]. Gender makes a difference according to Fteropoulli [11], but not according to Apers [4]. Severity of the CHD has been related to low QoL by a number of studies [7, 9–11], but a recent survey with a large sample size did not find this correlation [4]; moreover, Jackson [12] reported that patients suffering from moderate CHD lesions had the best QoL, in comparison to those with most severe conditions who had the worst QoL, and patients with mild abnormalities were in between those two groups.

QoL has been claimed to depend on many other factors beyond the clinicals and physicals [5], and this could explain some of those discrepancies. Personality traits [14] as well as social support [9, 10] may play a protective role. Cultural background, on the other hand, does not affect patients' QoL [4]. A recent study assessing the effect of standard of living and healthcare system characteristics, by means of national Gross Domestic Product per capita and total health expenditure per capita, showed that these variables do affect QoL [13]. However, this study used aggregated information, such as Gross Domestic Product per capita, and did not collect individual data, such as personal income, which allegedly may have an effect on these patients' QoL.

Regarding psychological wellbeing, the available evidence is also contradictory. Some studies report a higher rate of depression and anxiety, especially in patients with complex lesions [15], while other studies have more positive results, identifying a normal level of anxiety [3, 16] and hopelessness [17] in daily life. As was the case for QoL, socioeconomic factors have been related to anxiety and depression [18].

The primary aim of this study was to assess the QoL and psychological wellbeing of a population of adults suffering from CHD in Chile. In addition, it explores whether socioeconomic factors, such as poverty, are associated with the QoL of this population.

## Material and methods

### Design

This is a quantitative cross-sectional study. All procedures contributing to this work comply with the ethical standards of the relevant national guidelines on human experimentation (Law Nº20120, September 22nd 2006) and with the Helsinki Declaration of 1975, as revised in 2008, and have been approved by the institutional committees: Pontificia Universidad Católica de

Chile Faculty of Medicine committee and Hospital del Tórax committee. Patients agreed to participate and signed the informed consent form.

## Population

A group of adult patients with CHD who are being followed at the 'Instituto Nacional del Tórax', the main national reference center for Adult Congenital Heart Disease, were invited to participate. Inclusion criteria were: males or females over 18 years old with any type of congenital heart disease (repaired or unrepaired), a stable clinical condition (no recent acute decompensations or hospitalizations for surgical or percutaneous procedures). Exclusion criteria were: patients under 18 years old or unable to consent, illiterate, unwilling to participate or receiving more than two psychotropic drugs.

## Procedure

A researcher interviewed the participants in a private room while patients were waiting for a medical appointment. An informed consent form was signed beforehand and subjects were asked to complete a selection of psychological self-report scales and a socioeconomic survey questionnaire. The researcher remained available to provide clarification if needed. The scales and survey required 20 to 30 minutes approximately. Medical history was reported by patients and then compared with our local database.

## Measurement instruments

In order to assess QoL and wellbeing, the following questionnaires were used:

*The Medical Outcome Study 36-Item Short Form Health Survey (SF-36)* [19]. As suggested by Gill [20], before selecting a scale to assess QoL, it is important to define the meaning of QoL used in the study. We decided to use the definition coined by him, namely "a reflection of the way that patients perceive and react to their health status and to other, nonmedical aspects of their lives" [20] (p.619). Using this definition, we selected the Medical Outcome Study 36-Item Short Form Health Survey (SF-36). This scale is one of the most used to subjectively assess health status in biomedical studies. It contains 36 Likert questions and has 8 subscales: physical functioning, physical role functioning, bodily pain, general health perception, vitality, social role functioning, emotional role functioning, and mental health. Scores range from 0 to 100; higher scores indicate better subjective health status. The psychometric properties of this scale have been assessed in several countries and on different groups of patients, proving its accuracy. Although it does not consider all relevant dimensions to comprehensively assess QoL, it is the most used in this type of population [21]. The Spanish version was validated by Alonso [22].

*The General Health Questionnaire (GHQ-12)* [23]. This scale explores non-severe psychiatric symptoms in the general population and it evaluates mental health more than general health [24]. The selected version includes 12 Likert questions, scores between 0 and 36, and higher scores indicate worse mental health. It was validated in Spanish by Humphreys [25].

*The Basic Psychological Needs Scales (BPN)* [26]. The self-determination theory states that there are three universal psychological needs that must be met in order for people to experience psychological wellbeing, namely autonomy, competence and relatedness. They allow people to act according to their intrinsic motivations [27]. This scale assesses the degree to which these needs are met. The scale selected for this study was published by Gillet [26], and it consists of 15 items rated with a 5-point Likert scale, and scores from 15 to 75; higher scores indicate better fulfilment of psychological needs. The validated Spanish version has adequate internal consistency [28].

*The Beck Hopelessness Scale (BHS)* [29]. Hopelessness refers to a cognitive schema with negative expectations about the future. This scale is focused on pessimism and the belief that problems cannot be solved, both characteristics that could be related to depression. This is a 10-question true-or-false scale, and its results are expressed with a numeric value from 0 to 20, higher scores show more hopelessness; scores above 8 are considered abnormal and further assessment is advised. Some studies have assessed hopelessness in this population using the BHS [17]. The Spanish version was validated by Aguilar [30].

*Socioeconomic Level (SEL)*, an important variable to be taken into account, was measured by a sample of questions from the survey regularly conducted by the Chilean Ministry of Social Development [31]. The selected dimensions and indicators were: age, marital status, education (highest degree obtained), work status (employment, formal contract), health insurance, and housing/neighborhood (number of people living in the house, social participation). These socioeconomic data were analyzed independently and also transformed into a dichotomous variable (poverty) following the multidimensional poverty approach proposed by Aplablaza for the Chilean population [32]. This account has been used in a similar study [33].

*Clinical variables.* The severity of the disease was assessed using American College of Cardiology/American Heart Association guidelines [34]. Other variables included age and total number of past hospitalizations up to one year before the survey.

## Statistical analysis

Data on demographic variables and socioeconomic status was presented using descriptive parameters such as median and interquartile range because these variables' sample values had a non-normal distribution; data on QoL and psychological scales was shown using mean and standard deviation, as the distribution passed normal distribution tests. Associations between QoL and clinical and social measures, as well as with psychological scales, were analyzed with the Pearson or Spearman correlation, and t-tests. When the distribution did not pass normal distribution tests, the Kolmogorov-Smirnov test was applied. STATA v.13 (StataCorp, 4905 Lakeway Dr. College Station, TX 77845) was used to carry out the statistical analysis.

## Results

Data was collected from June to August 2019. During that period, 85 patients were contacted, 75 of whom were invited to participate as they met the inclusion criteria. 67 accepted and completed the questionnaires. The age of the participants ranged from 18 to 72 years old, median 29 (interquartile ranges (IQR) 22–38). 51.5% were female, 57.4% were single and 77.9% had completed secondary education. The total number of hospitalizations ranged from 1 to 30, median 2 (IQR 2–7) and 64.7% had a moderate CHD. More demographic characteristics of the participants are shown in Table 1. The results of the SF-36 and the psychological tests are shown in Table 2. 11 participants (16.2%) showed a hopelessness level considered abnormal (>8) and were referred for mental health evaluation at their local healthcare provider.

### Correlations between quality of life dimensions and clinical and psychological variables

Clinical dimensions did not show a significant correlation to any of the QoL dimensions. The Pearson's correlation coefficients range for these variables were: age from -0.22 to 0.19, education from -0.22 to 0.34, CHD severity from -0.3 to 0.13, and number of hospitalizations from -0.23 to -0.36.

On the other hand, psychological scales showed better correlation coefficients. GHQ had a good correlation to emotional and global dimensions (Pearson's coefficients from -0.45 to

**Table 1. Demographic data.**

| | |
|---|---|
| Age (years); median (IQR) | 29 (22–38) |
| Gender (female); n (%) | 35 (51.5) |
| Marital status; n (%) | |
| Married | 12 (17.7) |
| Cohabiting | 12 (17.7) |
| Separated | 2 (2.9) |
| Divorced | 2 (2.9) |
| Widowed | 1 (1.5) |
| Single | 39 (57.3) |
| Educational Level; n (%) | |
| Never attended | 0 (0) |
| Elementary | 0 (0) |
| Primary | 15 (22.1) |
| Secondary | 15 (22.1) |
| Secondary technical | 9 (13.2) |
| Technical | 20 (29.4) |
| Graduate | 8 (11.8) |
| Postgraduate | 1 (1.4) |
| Number of hospitalizations; median (IQR) | 3.5 (2–7) |
| Severity; n (%) | |
| Mild | 6 (11.8) |
| Moderate | 33 (64.7) |
| Severe | 12 (23.5) |
| Employment status (yes); n (%) | 34 (50.8) |
| Poverty (yes); n (%) | 40 (58.8) |

-0.7), BPN to almost all dimensions (Pearson's coefficients from 0.42 to 0.64), and hopelessness had a similar behavior to GHQ, showing a good correlation to emotional and global dimensions (Pearson's coefficients from -0.53 to -0.62). A complete correlogram is shown in Table 3.

**Table 2. Quality of life and psychological wellbeing.**

| | |
|---|---|
| SF-36; mean (SD) | |
| Physical function | 70.5 (26.4) |
| Physical role functioning | 64.0 (39) |
| Emotional role functioning | 77.8 (36.2) |
| Vitality | 65.3 (23.4) |
| Mental health | 72.7 (22.1) |
| Social role functioning | 75.4 (24.6) |
| Bodily pain | 72.1 (25.8) |
| General health perception | 58.6 (25.4) |
| GHQ-12; mean (SD) | 11.1 (8) |
| Basic psychological needs; mean (SD) | 64.7 (9.7) |
| Autonomy | 22.1 (3.6) |
| Relatedness | 21.8 (3.2) |
| Competence | 21 (3.5) |
| Beck Hopelessness Scale; mean (SD) | 4.1 (4.1) |

Quality of life and psychological scales results

**Table 3. Quality of life, clinical and psychological variables.**

| | Age | Education | Hospitalizations | Severity | GHQ-12 | BPN | Hopelessness | Physical function | Physical role functioning | Emotional role functioning | Vitality | Mental health | Social role functioning | General health |
|---|---|---|---|---|---|---|---|---|---|---|---|---|---|---|
| Age | 1 | | | | | | | | | | | | | |
| Education | -0.30 | 1 | | | | | | | | | | | | |
| Hospitalizations | 0.02 | -0.28 | 1 | | | | | | | | | | | |
| Severity | -0.46 | 0.06 | -0.04 | 1 | | | | | | | | | | |
| GHQ-12 | 0.19 | -0.20 | 0.23 | 0.10 | 1 | | | | | | | | | |
| BPN | -0.20 | 0.30 | -0.23 | -0.10 | **-0.60** | 1 | | | | | | | | |
| Hopelessness | 0.14 | -0.22 | 0.36 | 0.13 | **0.62** | **-0.63** | 1 | | | | | | | |
| Physical function | -0.22 | 0.34 | 0.01 | 0.04 | -0.24 | **0.44** | -0.28 | 1 | | | | | | |
| Physical role functioning | -0.05 | 0.15 | 0.02 | -0.23 | -0.18 | 0.35 | -0.25 | **0.77** | 1 | | | | | |
| Emotional role functioning | 0.04 | 0.05 | -0.06 | -0.30 | **-0.45** | 0.35 | -0.35 | 0.47 | **0.59** | 1 | | | | |
| Vitality | -0.15 | 0.20 | 0.01 | -0.25 | **-0.48** | **0.59** | **-0.53** | **0.55** | **0.56** | **0.60** | 1 | | | |
| Mental health | -0.11 | 0.03 | -0.05 | -0.15 | **-0.70** | **0.46** | **-0.56** | 0.32 | 0.34 | **0.61** | **0.72** | 1 | | |
| Social role functioning | -0.04 | 0.12 | -0.04 | -0.26 | **-0.61** | **0.42** | **-0.44** | **0.54** | **0.64** | **0.69** | **0.67** | **0.77** | 1 | |
| General health | -0.08 | 0.28 | -0.22 | -0.18 | **-0.56** | **0.64** | **-0.62** | **0.57** | **0.62** | **0.54** | **0.70** | **0.59** | **0.72** | 1 |

Correlations between quality of life dimensions, clinical and psychological variables

Values in bold: p<0.05

## Gender, poverty, quality of life and psychological wellbeing

Gender had a mild effect on QoL and no effect on psychological wellbeing, with the remarkable exception of physical function (males 82.66 and females 59.12; $p<0.01$) and physical role functioning (males 75 and females 53.68; $p<0.05$).

On the contrary, poverty was associated with low results in a number of the QoL dimensions, but without effect on psychological wellbeing. Poor participants scored worse than non-poor participants in physical function (61.92 versus 82.96; $p<0.01$), physical role functioning (53.21 versus 79.63; $p<0.01$), vitality (60.89 versus 71.67; $p<0.05$), social role functioning (70.19 versus 82.87; $p<0.05$) and bodily pain (65.77 versus 81.2; $p<0.05$). Results are provided in Tables 4 and 5.

## Discussion

As was commented beforehand, research in QoL studies shows ambiguous results [11, 21, 35]. This may be explained by heterogeneous measuring methods (e.g. study design, different populations, disease severity class, etc.) as well as a lack of methodological and conceptual rigor [11, 35]. QoL is a vague concept that opens up a large margin of interpretation [36, 37]. Diversion in perception and inappropriate use of the term QoL lead inevitably to inconclusive findings. With their review of QoL measurements, Gill and Feinstein set a milestone by developing 10 criteria that aim to support the evaluation of QoL measurements [35, 36]. Yet, more than 40 years later, there has been only a slight improvement in methodological and conceptual accuracy in QoL publications [21].

Despite this issue, our study has similar findings as some of those reported in the literature. We found a difference in QoL according to gender [11], and there was no association with CHD severity [4, 7, 13]. More important are the differences across the different dimensions of QoL. As other authors reported, CHD patients showed low scores in the physical domains of QoL, but the results in the social and emotional domains are comparable to those of the

**Table 4. Quality of life, psychological wellbeing and gender.**

|  | Female (n = 35) | Male (n = 32) |
|---|---|---|
| SF-36; mean (SD) |  |  |
| Physical function** | 59.12 (25.86) | 82.66 (21.4) |
| Physical role functioning* | 53.68 (43.58) | 75 (30.45) |
| Emotional role functioning | 75.49 (37.88) | 80.32 (34.73) |
| Vitality | 62.21 (23.75) | 68.59 (22.89) |
| Mental health | 69.76 (20.99) | 75.89 (23.11) |
| Social role functioning | 70.22 (25.93) | 80.86 (22.22) |
| Bodily pain | 66.25 (26.68) | 78.29 (23.71) |
| General health perception | 52.94 (25.49) | 64.53 (24.28) |
| GHQ-12; mean (SD) | 10.71 (7.28) | 11.48 (8.84) |
| Basic psychological needs; mean (SD) | 64.03 (10.86) | 65.36 (8.42) |
| Autonomy | 21.8 (4.16) | 22.39 (3.01) |
| Relatedness | 21.63 (3.65) | 21.97 (2.77) |
| Competence | 20.57 (4.98) | 21.55 (2.72) |
| Beck Hopelessness Scale; mean (SD) | 3.76 (3.23) | 4.36 (5.01) |

Quality of life and psychological scales results according to gender.

*: $p<0.05$

**: $p<0.01$

**Table 5. Quality of life, psychological wellbeing and poverty.**

|  | Non-poor (n = 28) | Poor (n =) 40 |
|---|---|---|
| SF-36; mean (SD) |  |  |
| Physical function** | 82.96 (17.88) | 61.92 (28.11) |
| Physical role functioning ** | 79.63 (29.45) | 53.21 (41.43) |
| Emotional role functioning | 84.08 (29.7) | 73.51 (39.87) |
| Vitality* | 71.67 (21.97) | 60.89 (23.56) |
| Mental health | 75.26 (23.38) | 70.97 (21.28) |
| Social role functioning* | 82.87 (21.69) | 70.19 (25.43) |
| Bodily pain* | 81.2 (22.42) | 65.77 (26.38) |
| General health perception | 64.81 (25.63) | 54.23 (24.64) |
| GHQ-12; mean (SD) | 12.43 (9.47) | 10.15 (6.81) |
| Basic psychological needs; mean (SD) | 63.79 (8.79) | 65.3 (10.36) |
| Autonomy | 21.89 (3.24) | 22.23 (3.92) |
| Relatedness | 21.39 (2.78) | 22.08 (3.53) |
| Competence | 21.14 (2.86) | 20.98 (3.91) |
| Beck Hopelessness Scale; mean (SD) | 4.86 (5.43) | 3.49 (2.93) |

Quality of life and psychological scales results according to poverty.

*: $p < 0.05$

**: $p < 0.01$

general population [5, 11, 38]. Our findings support the notion that QoL is a complex concept, which is related to factors beyond physical functioning and clinical facts (such as age, CHD severity or number of hospitalizations) [4, 5, 13], highlighting the importance of social support [10]. Despite the fact that an association between CHD and socioeconomic burden has been described [39], this variable is frequently neglected when surveying this population's QoL. Aiming at assessing this factor, Jackson et al. surveyed individual income of CHD patients and showed that earning less than US$30,000 per year explained 23% of the variability in the QoL [12]. A recent study in Chinese children suffering from CHD showed that there is an association between QoL and socioeconomic status [40], assessed by means of household income, parental occupation and educational level. Nevertheless, the mechanism by which socioeconomic level influences QoL is still not fully characterized. It has been suggested that CHD causes poverty or that inability to pay makes access to healthcare more problematic [12], which is not the case for Chilean patients. Other plausible explanations are an eventual link between poverty and low health literacy, affecting patients' adherence and healthcare consulting behavior [40], and the effect of long-term and comprehensive rehabilitation therapy, which is often expensive. The association between socioeconomic status and these patients' QoL should be carefully explored, as well as the underlying mechanisms. This population has an increased risk of suffering from economic hardships [39] and any effort to improve their QoL that neglects socioeconomic factors would be rather ineffective.

QoL includes different areas, and the subjective perceived QoL may be associated with psychological wellbeing. An appropriate psychological functioning is crucial to all human beings. As was the case for QoL, several studies with CHD patients have been conducted assessing psychological wellbeing, revealing inconsistent findings. Whereas some studies describe higher rates of anxiety and depression [15, 38, 41], others report no differences compared to healthy controls [3, 16, 17] and outline independent variables like social support and socio-economic factors that affect people's level of anxiety and depression [18]. The importance of the individual psychological wellbeing is highlighted by the fact that it influences the adherence to

medical treatment and hence affects patient's recovery [42]. Moreover, poor emotional (and physical) QoL might result in struggles with important tasks like physical activities or attending medical appointments [12]. In order to evaluate the relation between QoL and psychological wellbeing of our study patients, we made use of the GHQ-12 and the BPN scale. Results from the GHQ-12 questionnaire revealed a good negative correlation with different QoL factors like vitality, social functioning and general health, whereas BPN showed positive correlations to almost all dimensions. Neither gender nor poverty had a significant effect on psychological functioning. This is a very interesting finding, as arguably both issues have been related to low psychological wellbeing [30]. Patients suffering from CHD, however, are a particular population, and the psychological resources they use to face their disease might be applied to cope with gender and socioeconomic issues as well. This explanation should be further explored.

Providing the appropriate care to these patients is still a challenge. Diagnosis and medical treatment frequently puzzle healthcare teams, and structural abnormalities are often difficult to repair. Despite these issues, mortality has importantly decreased and, according to our results, psychological wellbeing is also preserved. However, if we want to offer these patients good QoL, attention should be paid to those factors that compromise physical functioning and global QoL. Our study suggests that socioeconomic factors need to be taken into account.

Our study has many limitations that have to be acknowledged. The sample size is small and we recruited patients from only one public center. However, the 'Instituto Nacional del Tórax' is the main reference center treating these patients, and receives patients from the whole country. This fact, added to a high response rate (89.3%), enhanced our outcomes' representativeness. Having a control group and comparing the results of both populations would have provided very useful information to have a better understanding of the phenomenon. We intended to extend the project in this way, but the sociopolitical events that happened in Chile from October 18th 2019 onwards and the COVID-19 pandemic thwarted our intention, as the levels of anxiety and hopelessness skyrocketed in the Chilean population and the results would not have been comparable.

The proportion of patients classified as poor is remarkably high (58.8%). This fact could be considered as a limitation, due to lack of representativeness. However, in this study we surveyed socioeconomic factors using a multidimensional approach, considering dimensions beyond household income, such as education, working status, health insurance and housing, with specific cutoff values, according to Chilean standards [32]. Multidimensional poverty indexes usually classify more people under the line of poverty than pure income approaches, but they provide a well-grounded method to identify people suffering from socioeconomic deprivation, rendering them vulnerable. Furthermore, this possible overrepresentation of the poor does not challenge the differences found when comparing this group with the non-poor.

Regarding the methodological issues that have been identified in the studies exploring QoL of CHD patients, our study meets a number of the rigor criteria, and those that were not fulfilled were left as such after extensive discussions.

Despite the relevance of this topic, and the huge amount of available literature, there is not much evidence regarding QoL and the psychological wellbeing of adults with CHD in Latin America. To our knowledge, this is the first study exploring QoL and the psychological wellbeing of Chilean adult CHD patients.

In conclusion, this study provides evidence that poverty is associated to low QoL in Chilean patients suffering from CHD. These patients show low QoL in the physical dimensions, but this phenomenon is not seen in other QoL dimensions, suggesting that it depends on factors beyond physical functioning and clinical tests. Further studies should be done in order to have a better understanding of this phenomenon, such as longitudinal studies and qualitative

research. This knowledge will allow us to design effective strategies, aiming at improving the QoL for this vulnerable population.

## Acknowledgments

The authors wish to thank Margarita Bernales PhD, Guillermo Lema MD and Justine Robertson for their invaluable comments and suggestions during the writing of this manuscript.

## Author Contributions

**Conceptualization:** Rodrigo López Barreda, Manuela Scotoni, Fernando Baraona.

**Data curation:** Rodrigo López Barreda, Alonso Guerrero, Juan Cristóbal de la Cuadra, Manuela Scotoni, Fernando Baraona.

**Formal analysis:** Rodrigo López Barreda, Manuela Scotoni, Wilbaldo Salas, Francisca Arancibia.

**Investigation:** Rodrigo López Barreda, Alonso Guerrero, Juan Cristóbal de la Cuadra, Fernando Baraona.

**Methodology:** Rodrigo López Barreda, Manuela Scotoni, Wilbaldo Salas.

**Project administration:** Rodrigo López Barreda.

**Supervision:** Rodrigo López Barreda, Fernando Baraona, Polentzi Uriarte.

**Writing – original draft:** Rodrigo López Barreda, Alonso Guerrero, Juan Cristóbal de la Cuadra, Manuela Scotoni, Wilbaldo Salas, Francisca Arancibia, Polentzi Uriarte.

**Writing – review & editing:** Rodrigo López Barreda, Manuela Scotoni, Wilbaldo Salas, Fernando Baraona, Francisca Arancibia, Polentzi Uriarte.

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
