## [Decision Letter · Decision Letter 0]

27 Aug 2020

PONE-D-20-19646

Poverty and Quality of Life and Psychological Wellbeing in Adults with Congenital Heart Disease in Chile

PLOS ONE

Dear Dr. Lopez Barreda,

Thank you for submitting your manuscript to PLOS ONE. After careful consideration, we feel that it has merit but does not fully meet PLOS ONE’s publication criteria as it currently stands. Therefore, we invite you to submit a revised version of the manuscript that addresses the points raised during the review process.

Your submission is of interest to the readership as indicated by the reviewer. However, the reviewer also raised the concern in the presentation of the manuscript. Therefore, please invite a native English speaker who has experience in academic writing to help you polish the presentation before your resubmission. Thank you.

We look forward to receiving your revised manuscript.

Kind regards,

Chung-Ying Lin

Academic Editor

PLOS ONE

Journal Requirements:

3.We note that you have indicated that data from this study are available upon request. PLOS only allows data to be available upon request if there are legal or ethical restrictions on sharing data publicly. For information on unacceptable data access restrictions, please see http://journals.plos.org/plosone/s/data-availability#loc-unacceptable-data-access-restrictions.

Reviewers' comments:

Reviewer's Responses to Questions

**Comments to the Author**

1. Is the manuscript technically sound, and do the data support the conclusions?

Reviewer #1: Yes

2. Has the statistical analysis been performed appropriately and rigorously? 

Reviewer #1: Yes

3. Have the authors made all data underlying the findings in their manuscript fully available?

Reviewer #1: Yes

4. Is the manuscript presented in an intelligible fashion and written in standard English?

Reviewer #1: No

5. Review Comments to the Author

Reviewer #1: I appreciate the author's work and dedication to elucidate the relationship among psychological wellbeing and quality of life among adults with complex congenital heart disease.

- The background is thorough and organized well.

- Given extant research is growing and inconclusive, I would like to see this manuscript take a firmer stance on their hypotheses and rationale

- The discussion is well written and comprehensive. Perhaps more of a discussion on the poverty piece (e.g., are patients with CHD more at risk for being in poverty)?

- This manuscript would benefit from a clear paragraph on (1) what this study has added to the literature that exists and (2) what future research should aim to better understand

6. PLOS authors have the option to publish the peer review history of their article (what does this mean?). If published, this will include your full peer review and any attached files.

Reviewer #1: No

---

## [Author Response · Author response to Decision Letter 0]

23 Sep 2020

A major linguistic review was undertaken by a native speaker.

The introduction was reshaped, making clearer the authors’ stance on the topic, what we expected to find and the reasons behind this hypothesis.

The relationship between CHD and poverty was further commented and one new reference was used to illustrate this complex interaction.

One paragraph of the discussion was rephrased to emphasize our contribution to current literature and possible research projects to advance in the knowledge of this issue.

---

## [Editor Report · Decision Letter 1]

25 Sep 2020

Poverty, quality of life and psychological wellbeing in adults with congenital heart disease in Chile

PONE-D-20-19646R1

Dear Dr. Lopez Barreda,

We’re pleased to inform you that your manuscript has been judged scientifically suitable for publication and will be formally accepted for publication once it meets all outstanding technical requirements.

Kind regards,

Chung-Ying Lin

Academic Editor

PLOS ONE
---

## [Editor Report · Acceptance letter]

28 Sep 2020

PONE-D-20-19646R1 

Poverty, quality of life and psychological wellbeing in adults with congenital heart disease in Chile 

Dear Dr. López Barreda:

I'm pleased to inform you that your manuscript has been deemed suitable for publication in PLOS ONE. Congratulations! Your manuscript is now with our production department. 

Kind regards, 

on behalf of

Dr. Chung-Ying Lin 

Academic Editor

PLOS ONE